# Playing Behavior Classification of Group-Housed Pigs Using a Deep CNN-LSTM Network

**Beng Ern Low** [1] , **Yesung Cho** [1] , **Bumho Lee** [1] **and Mun Yong Yi** [2,*]

1   Graduate School of Data Science, Korea Advanced Institute of Science and Technology (KAIST), Daejeon 34141, Republic of Korea
2   Department of Industrial and Systems Engineering, Korea Advanced Institute of Science and Technology (KAIST), Daejeon 34141, Republic of Korea
*   Correspondence: munyi@kaist.ac.kr

**Abstract:** The swine industry is one of the industries that progressively incorporates smart livestock farming (SLF) to monitor the grouped-housed pigs' welfare. In recent years, pigs' positive welfare has gained much attention. One of the evident behavioral indicators of positive welfare is playing behaviors. However, playing behavior is spontaneous and temporary, which makes the detection of playing behaviors difficult. The most direct method to monitor the pigs' behaviors is a video surveillance system, for which no comprehensive classification framework exists. In this work, we develop a comprehensive pig playing behavior classification framework and build a new video-based classification model of pig playing behaviors using deep learning. We base our deep learning framework on an end-to-end trainable CNN-LSTM network, with ResNet34 as the CNN backbone model. With its high classification accuracy of over 92% and superior performances over the existing models, our proposed model highlights the importance of applying the global maximum pooling method on the CNN final layer's feature map and leveraging a temporal attention layer as an input to the fully connected layer for final prediction. Our work has direct implications on advancing the welfare assessment of group-housed pigs and the current practice of SLF.

**Keywords:** animal positive welfare; pig play behavior; convolutional neural network; long short-term memory network; video classification





## 1. Introduction

Along with the emergence of the Fourth Industrial Revolution, smart livestock farming (SLF) is being considered as a realistic way to effectively meet the global food demand. SLF is an emerging concept that seeks to improve livestock farming conditions by effectively utilizing digital technologies including the real-time supervision of the behaviors of livestock. Due to the SLF tools' real-time and non-invasive nature, swine practitioners generally use them to closely monitor pigs' behavior, welfare, and growth [1,2]. Among these tasks, behavior and welfare monitoring has gradually gained much attention these days, as there is an increasing awareness and demand from global consumers for a better farming environment for piglets in food production as standard practice [3].

As suggested by [4,5], qualitative and quantitative measurement of behavioral expression is essential in identifying and facilitating the conditions conducive to positive welfare. Accordingly, within the animal behavioral research community, instead of just focusing only on pigs' negative problems, there has been an increasing number of studies related to behavioral indicators of pigs' positive welfare, such as social affiliative behaviors [6–10] and play behaviors [11–19].

Pig play behaviors are evident natural indicators that positive welfare exists in a particular pigpen. Animals only express their play behaviors when they are under a favorable and non-life-threatening environment [4,20,21]. Not only do play behaviors indicate pleasant experiences, but they could also contribute to the development of physical, cognitive,

and affective competence, thereby promoting the positive welfare state [21]. In the work of [22], they found that pig playing behaviors were associated positively with physical development, especially the weight change between birth and weaning of piglets. In addition, several studies showed how play behavior is associated with positive welfare conditions such as low ammonia [23] or an enriched environment [24].

Undoubtedly, it is practically impossible for pig farmers to directly monitor and analyze the group-housed pig behaviors on-farm twenty-four seven. Fortunately, the advancement of affordable video surveillance technology has made continuous monitoring of pig behavior practically possible. Nonetheless, it is labor-intensive and time-consuming to train a pig farmer to continuously monitor and analyze the pig behavior from the video display. Therefore, there is a growing number of studies in computer vision to automatically recognize diverse pig behaviors from the video by developing an artificial intelligence (AI) model using a huge volume of collected video data [25,26].

Following the success of deep learning (DL) in the computer vision field, particularly the convolutional neural networks (CNNs), the methodology of pig behavior recognition has shifted from the conventional computer vision to DL. Not only have CNNs been widely applied by researchers in the livestock sector but also by researchers in various fields, including the medical field [27,28] and structural engineering field [29,30]. Most of the prior studies focused on recognizing pig behaviors such as feeding, drinking, mounting, aggressive, and postural behaviors, as well as multiple locomotive movements. In contrast, detecting pig playing behaviors has been much less studied, primarily due to data scarcity.

Recent studies that focused on recognizing pig playing behaviors from the video clips mostly relied on a deep neural network (DNN) that consisted of a CNN and long short-term memory (LSTM), which is also known as a CNN-LSTM network. For example, ref. [31] sought to recognize the object play or engagement behaviors of fattening pigs with different enrichment objects by first tracking the enrichment object and then locating the region of interest (ROI) that includes the target pig and the enrichment object. The final circular ROI was constructed by setting the centroid of the detected enrichment object as the circle center and the average pig length (220 pixels) as the radius. Each episode frame with pixels other than the ROI removed was consequently fed into a CNN-LSTM network for the downstream behavior recognition task. The CNN backbone used in this work is InceptionV3 [32]. Another set of experiments was also carried out with the radius of circular ROI shortened into half to investigate the effect of ROI on model performance [33]. It is worth mentioning that in the work of [34], they also used a CNN-LSTM network with VGG16 [35] as the CNN backbone to recognize the aggressive episodes at the group level within a pigpen. The model was trained to detect the occurrence of aggressive behaviors at the pen level instead of the individual level. All the video clips or episodes used in the aforementioned work contained a much smaller number of pigs (fewer than 15) and a simpler or more consistent background than in our work; hence, they could achieve good performance by simply applying the original CNN-LSTM architecture. None of them attempted to alter the CNN-LSTM architecture to improve the model performance.

In this study, we propose a deep-learning-based framework that can recognize pig playing and non-playing behaviors, covering a comprehensive range of playing behaviors from the video episodes, where their occurrences can serve as an indicator of assessing positive welfare for a commercial pig production system. Traditional machine learning methods require feature engineering involving human experts. Our study aims at detecting pigs' playful behaviors without feature engineering. To the best of our knowledge, this study represents the first endeavor to include three pig playing behavior categories, namely, social play, object play, and locomotor play for comprehensive pig playing and non-playing behavior classification. In particular, we develop an end-to-end trainable deep neural network that can work well in a more complex situation involving a complicated background and context, and a large number of pigs (50–60 per pigpen). To adapt the original CNN-SLTM architecture to this environment, we propose a few significant modifications to the existing CNN-LSTM network to boost the model performance, including the

global maximum pooling method and temporal attention layer. Our approach does not require any feature engineering, and it has direct implications on the welfare assessment of group-housed pigs and the current practice of SLF.

## 2. Materials and Methods

### 2.1. Datasets

2.1.1. Data Acquisition

Data used in this study were collected directly from a commercial pig farm in Hampyeong-gun, Jeollanam-do, South Korea, in collaboration with Animal Industry Data Korea (AIDK), a livestock healthcare solution provider company based in South Korea. The nursery pigs, aged between 25 to 50 days old, were housed and monitored closely in 2 separate pig pens, and each pigpen contained 50 to 60 pigs. Each pen had a size of 3.6 m (V) × 3 m (W) × 8 m (H). A feed stand was placed at the center of each pen and a bucket was placed at its corner to store materials to manage excrements. There was an irregular arrangement of the toys (e.g., rubber balls, wooden sticks, and rubber tires) to help the pigs relieve their stress by playing with them. The pens typically get cleaned once or twice a month depending on the season.

To collect the video data, a top-view camera (CCTV) was installed in the middle of each nursery pigpen, perpendicularly above the pigpen floor. A normal lens camera was fixed with a focal distance of 4 mm and a relative aperture of F1.4. The angular field of view is set as 88.6° (H) × 47.5° (V) × 104.8° (D). Cameras are adjusted to have fixed focus control, with a minimum focusing distance of 0.5 m. Each pigpen was recorded for 24 h at a rate of 30 frames per second, with a resolution of 1920 × 1080. The pigpen was recorded continuously from 15 November 2021 to 29 December 2021.

For this study, four 1 min video clips were selected for each pigpen, at 7.30 a.m., 11 a.m., 6 p.m., and 11 p.m. The collection schedule was set based on the work of [36], which analyzed the videos that captured the locomotive behaviors of pigs in the morning, afternoon, and evening. We added 11 p.m. to expand the coverage and include nighttime behaviors. The 1 min length of clips was determined based on the work of [37], who found that 1 min assessments were sufficient to distinguish two groups of pigs based on behavior.

Each 1 min video clip (Figure 1) came with a JSON file, which contained the growth phase, ID, and the corresponding bounding box coordinates of each of the pigs that appear in the video frame. Particularly, we used video clips that were collected after the nursery pigs were regrouped for more than 10 days to avoid the hierarchy establishment period [38] that involves aggressive fighting due to the mixing of unfamiliar pigs right after weaning [39,40]. A total of 72 1 min video clips were eventually obtained for our work.

2.1.2. Data Preprocessing

To enhance the coverage of bounding boxes and extract the behavior of each pig in the scene, we applied an additional preprocess procedure. We made use of the bounding box coordinates and pig ID to crop and extract the episodes from the 1 min video clips for each pig, with the target pig centered in the frame. One limitation we have with this method of automated episode extraction based on the bounding box is that the bounding box of certain pigs is occasionally missing when the pigs are inactive or have very minor movements, overlapped with other pigs, or covered by other objects such as pigpen poles and feeders. Hence, we employed a very simple and straightforward bounding box imputation method to harness more episodes, which is particularly useful for pigs that show minor movement. Whenever an 'empty' bounding box is detected for a pig ID in each frame, we check if the nearest 45 frames contain any non-empty bounding box. If there exists non-empty bounding box information from the nearest 45 frames, we impute the missing bounding box using the coordinates from the nearest frame.

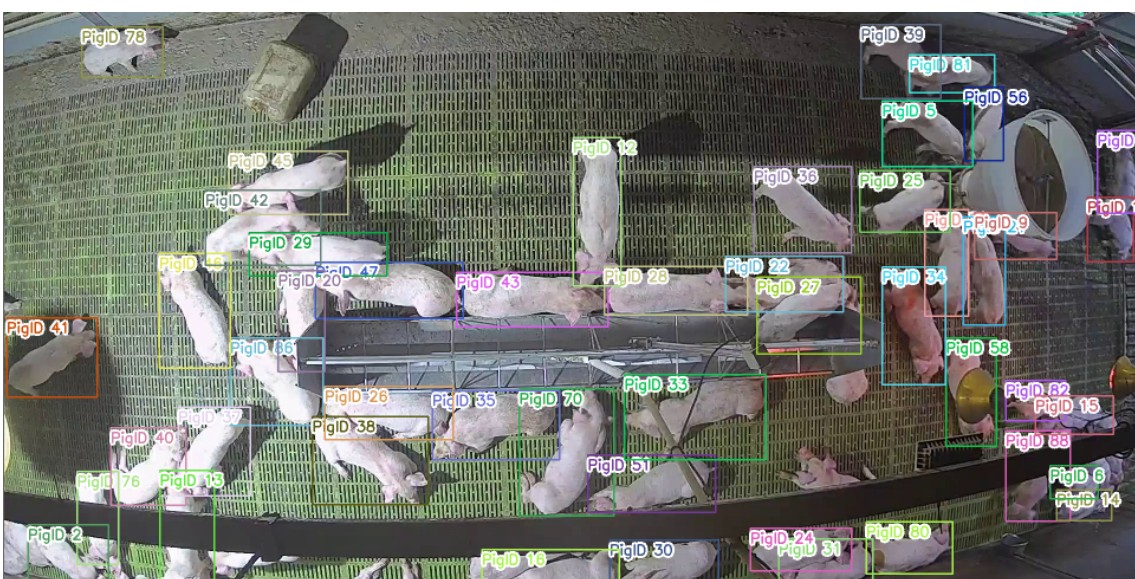

**Figure 1.** A sample frame of video clips recorded in the nursery pigpen, with bounding box and ID information.

After imputation, the maximum width and height of the bounding box for each pig ID in the 1 min clip were obtained. To extract the bounding box with consistent width and height for each pig ID across all frames in each video clip, we expanded the biggest bounding box of each pig ID by a factor of 1.2 to obtain the new bounding box coordinates for further episode extraction. The rationale for expanding the bounding box size is to capture the context, such as interactions with other pigs, feeders, enrichment objects, and pen facilities around the target pig. Hence, after a few preliminary extraction attempts with different expansion factors, the factor of 1.2 was eventually decided since it gave minimal noise and optimal area for context understanding from a human observer's point of view. Then, for each pig ID, we extracted the cropped episodes based on the new bounding box coordinates.

To increase the dataset size and the efficiency of the labeling task, we divided and trimmed all the 1 min clips extracted for each pig equally into multiple 2 s episodes. We selected the fixed episode length of 2 s instead of 1 s, following the approach from previous work [33] in order to identify the noticeable aggressive behaviors. For instance, when a pig approaches another pig's ear, we cannot typically identify if the pig is biting (aggressive) or just nudging (playing) in just one second. However, in a 2 s time window, it is possible for us to observe whether the receiver pig shows an unpleasant reaction, which indicates whether the receiver pig's ear is bitten or nudged [41]. In the process of preliminary data screening, 2 s episodes that appeared to be shaky due to the unstable bounding box coordinates or with fewer than 60 frames were discarded.

### 2.1.3. Data Labeling

Video clips were manually labeled following the ethogram developed in [8,13–15,22,42] and the advice from professional veterinarians. There are 3 main categories of pig playing behavior, namely social play, object play, and locomotor play. As for the remainder of the paper, a pig's behavior is categorized as a playing behavior following the ethogram that contains 17 fine-grained playing behaviors (see Table A1 under Appendix A). If the target pig showed a playing behavior for at least 1 s in a 2 s episode, the episode is labeled as playing, and vice versa.

By strictly following the ethogram and 1 s rule mentioned above, the main researcher first manually labeled all episodes' binary play class, play category, and dominant playing behaviors. Two other researchers then manually labeled the episodes separately and independently after going over the guideline and observing a one-time demonstration

of the labeling by the main researcher. They inspected a total of 4466 episodes and initially disagreed on the labels of 524 episodes, which account for about 11.7% for the four categories (social, object, locomotor, and non-play). To ensure the objectivity of the class labels, a face-to-face discussion was conducted to resolve the conflict in the combined labeling results. The episodes were re-labeled with the ultimate objective of achieving unanimity. The episodes were discarded when a unanimous agreement could not be achieved after discussion. The whole labeling plus post-agreement discussion process took approximately 60 h to finish. After the intensive data labeling process, a total of 4333 2 s episodes were finally obtained. Table 1 depicts the distribution of the labeled episodes.

**Table 1.** Breakdown of Playing and Non-Playing Episodes.

| Class | Play Category | Number of Episodes (Ratio) | Total | Ratio |
|---|---|---|---|---|
| Playing | Social | 825 (0.1904) | 1029 | 0.24 |
| | Object | 178 (0.0411) | | |
| | Locomotor | 26 (0.0060) | | |
| Non-playing | - | 3304 (0.7625) | 3304 | 0.76 |
| | | | 4333 | 1.00 |

2.1.4. Train–Test Set Creation

Before splitting the final dataset of 4333 episodes into the train set and test set, we manually selected 203 episodes as Test Set 2 for model robustness testing. The episodes in Test Set 2 possess the following characteristics:

- From the top-view angle, the target pig's body is partially blocked by the objects such as a pole, feeder, and lamp for at least 1 s.
- The target pig is not located in the middle of the video frame.
- When the target pig is captured near the feeder, the feeder appears proportionally larger than the target pig.

By comparing the sample frames taken from Train Set, Test Set 1 (Figure 2), and Test Set 2 (Figure 3), we can see that Train Set and Test Set 1 contain episodes with less noise (obstacles) and with the target placed in the middle. The remaining 4130 episodes were split into Train Set and Test Set 1 with a ratio of 8:2 by stratified random sampling to retain the ratio of playing to non-playing episodes. Table 2 presents a summary of our train and test data. Additionally, to mitigate the imbalanced dataset problem, we performed weighted oversampling during model training, where we assigned higher weights to the playing episodes and lower weights to non-playing episodes. After applying the weighted oversampling strategy, the total number of episodes used during the training was 5028 (2514 playing and 2514 non-playing).

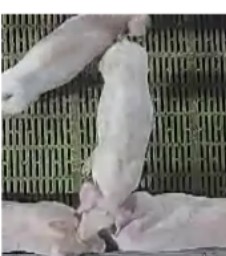 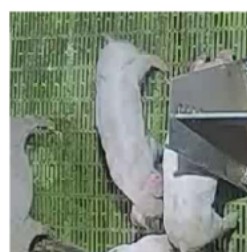 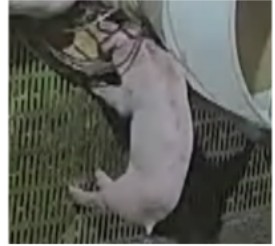 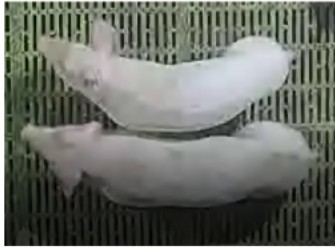

**Figure 2.** The sample frames of Train Set and Test Set 1 for model training and testing.

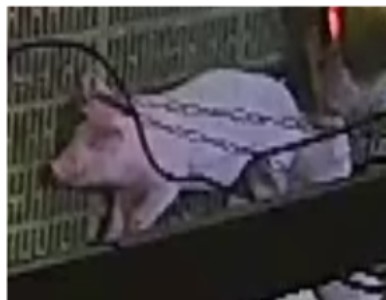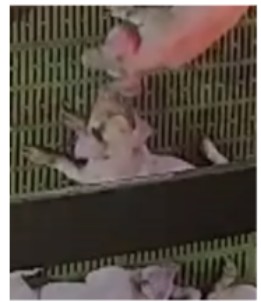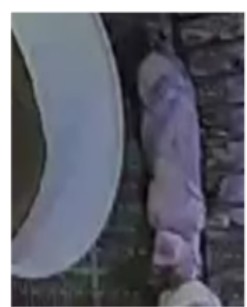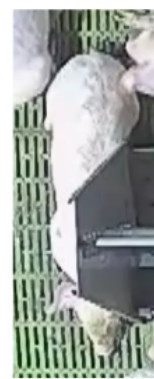

**Figure 3.** The Sample Frames of Test Set 2 for Model Robustness Testing.

**Table 2.** Distribution of Train Set, Test Set 1, and Test Set 2. The Numbers in the Brackets Represent the Ratio of the Class.

|  | Train Set | Test Set 1 | Test Set 2 |
|---|---|---|---|
| Non-playing | 2514 (0.76) | 628 (0.76) | 162 (0.80) |
| Playing | 790 (0.24) | 198 (0.24) | 41 (0.20) |
| Total | 3304 | 826 | 203 |

### 2.2. Methodology

#### 2.2.1. Algorithm

Our proposed deep learning network in this study was based on a deep CNN-LSTM network, as depicted in Figure 4. A ResNet34 [43] model pretrained on ImageNet [44] and a single-layer LSTM [45] network containing 60 hidden units were used to form the CNN-LSTM network. By treating each frame from the preprocessed 2 s episodes as an input image, the frame was first resized to $256 \times 256$ and then center-cropped to a size of $224 \times 224$, following the input image requirement of ResNet34.

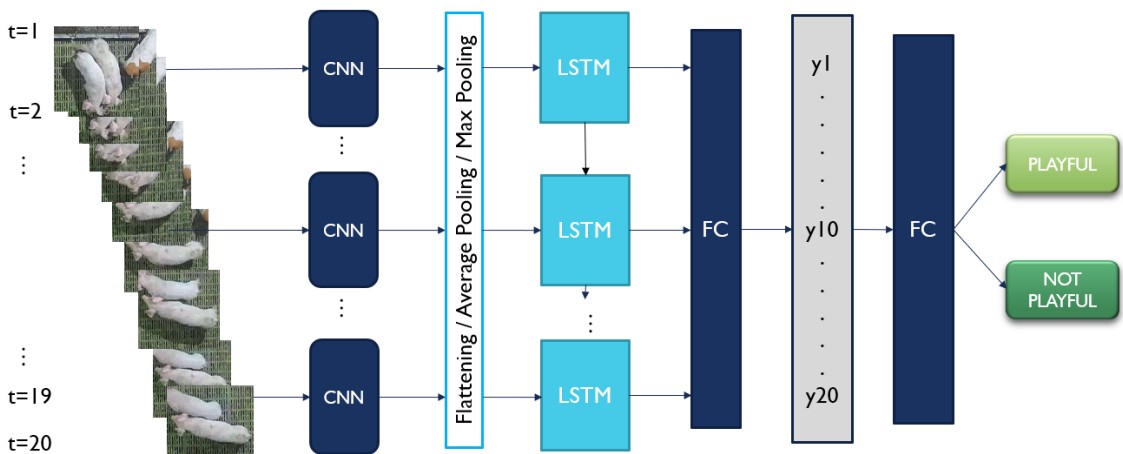

**Figure 4.** CNN-LSTM network (baseline/original variant) used in our work. FC = fully connected layer.

Inspired by the results of [34], we sampled only 20 equidistant frames instead of all 60 frames from each episode and fed them into the backbone ResNet34 model frame by frame to extract the spatial feature. The noticeable drawbacks of feeding all frames to the model such as [31,33] are redundant visual and motion information and higher computational cost. The extracted spatial feature acted as an input to the LSTM network.

The hidden state extracted by the LSTM for each frame was treated as a spatio-temporal feature. The hidden state output was then fed into a fully connected layer and a Softmax layer, which eventually yielded two probabilities of both playing and non-playing classes, respectively. The episode was predicted as the class with a higher probability.

### 2.2.2. CNN Spatial Feature Map

To convert the spatial feature maps ($7 \times 7 \times 512$ dimensions) extracted from the last layer of ResNet34 to the input feature of LSTM, we experimented with 3 alternative operations on the feature maps (See Figure 4):

1.  Flattening: The feature maps extracted from the last layer of ResNet34 model were simply flattened and concatenated to obtain a fixed-length vector representation (25,088-dimensional input feature of LSTM).
2.  Global average pooling: Global average pooling was employed to downsample the feature maps extracted from the last layer of ResNet34 model, in which the average value of all pixels in each feature map was concatenated to be a fixed-length vector representation (512-dimensional input feature of LSTM). As compared with global maximum pooling, it gives a more general representation of each feature map.
3.  Global maximum pooling: Global maximum pooling was used to downsample the feature maps extracted from the last layer of the ResNet34 model, in which the pixel with the highest value (brightest pixel) of each feature map was concatenated to be a fixed-length vector representation (512-dimensional input feature of LSTM). As compared with global average pooling, it produces the most important representation of each feature map, which in our case is the pigs that are white in color.

### 2.2.3. LSTM Classifier

In addition to the conventional prediction approach of a CNN-LSTM network (See Figure 4), we experimented with 3 alternative methods (hereinafter collectively referred to as LSTM classifiers) of dealing with the spatio-temporal feature extracted from LSTM to make the final prediction (See Figure 5):

1.  Last hidden state: LSTM is well-known for its ability to learn the temporal information of a long sequence progressively. In a time series prediction task using a deep neural network, the prediction is usually made using the final hidden state (latent representation learned) extracted. We applied the same concept by feeding the last spatio-temporal feature vector extracted by LSTM to the fully connected layer.
2.  All hidden states: In this approach, spatio-temporal feature vectors extracted by LSTM from all video frames were fused by simple concatenation. The concatenated vector was then passed to the fully connected layer.
3.  All hidden states with temporal attention: In this approach, spatio-temporal feature vectors, $\mathbf{h}_t^s$, extracted by LSTM from all video frames were attended differently. A temporal attention layer (Equations (1) and (2)) was added after concatenation of the spatio-temporal feature vectors. Then, the final representation vector ($a^s$ derived from Equation (3)) formed was passed to the fully connected layer.

$$\tilde{\alpha}_t^s = u_\alpha^T \tanh(W_\alpha h_t^s + b_\alpha) \tag{1}$$

$$\alpha_t^s = \frac{\exp \tilde{\alpha}_t^s}{\sum\limits_{t=1}^{20} \exp \tilde{\alpha}_t^s} \tag{2}$$

$$a^s = [\alpha_1^s h_1^s, \ \alpha_2^s h_2^s, \ ..., \ \alpha_{20}^s h_{20}^s] \tag{3}$$

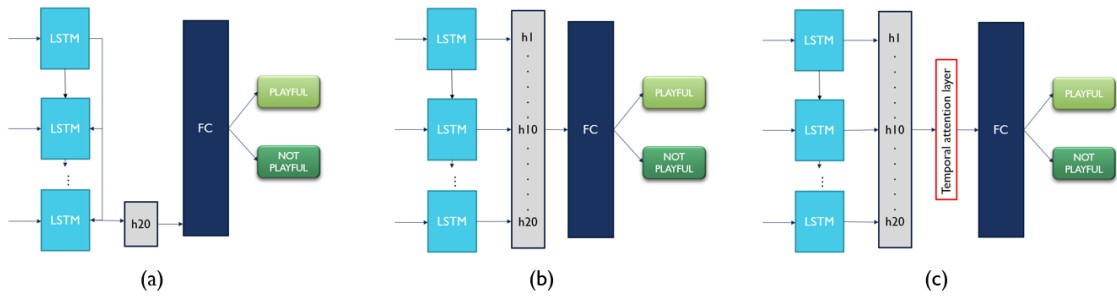

**Figure 5.** The 3 alternative approaches to make final prediction based on the extracted spatio-temporal feature from LSTM: (**a**) Last hidden state, (**b**) All hidden states, (**c**) All hidden states with temporal attention. $h_n$ stands for $n$th hidden state. FC stands for fully connected layer.

### 2.2.4. Experiment Implementation Details

All models were trained on an NVIDIA GeForce RTX 3080i GPU with 10 GB of memory. We implemented our DNN models using libraries and packages from PyTorch [46] written in Python programming language. For video reading, writing, and cropping, we used OpenCV library [47] written in the Python programming language. To visualize the CNN backbone's final layer's localized activation map, we adopted the Gradient-weighted Class Activation Mapping (Grad-CAM) method [48] by using the PyTorch library provided by [49].

In all our experiments, due to the GPU memory constraint, we used a batch size of 4 with stochastic gradient descent (SGD) as the optimizer. The initial learning rate was set as 0.01, and the learning rate was decreased by a factor of 0.2 when the accuracy of Test Set 1 did not improve for 5 epochs. An early stopping strategy was also implemented to stop the training process when the loss did not become decreased for 20 epochs. Using the CNN-LSTM network variant with only global average pooling, four separate experiments were conducted to obtain the optimal number of hidden units of LSTM. After training and evaluating the baseline model on 30, 60, 90, and 100 hidden units, we found that 60 is the optimal number of hidden units for our case. A summary of hyperparameter settings is presented in Table 3.

**Table 3.** Summary of Hyperparameter Settings.

| Parameter | Value |
|---|---|
| Batch size | 4 |
| Optimizer | SGC |
| Learning rate | 0.01 with scheduler |
| Input size | $224 \times 224$ |
| Number of hidden units | 60 |

Our loss was calculated using the binary cross-entropy loss function. For each batch, the loss $l_B$ is calculated as per Equation (4), where $B$ is the batch size. The loss per epoch $l_N$ is calculated as per Equation (5), where $N$ is the size of the train or test set.

$$l_B = \frac{\sum_{n=1}^{B} l_n}{B}, \text{ where} \tag{4}$$

$$l_n = -(y_n \cdot \log x_n + (1 - y_n) \cdot \log(1 - x_n))$$

$x_n$: predicted probability of nth sample;
$y_n$: ground truth of nth sample, where 1 = playing behavior, 0 = non-playing behavior.

$$l_N = \frac{\sum\limits_{n=1}^{N} l_n}{N} \tag{5}$$

As our research task is a supervised classification task, we use the standard binary classification metrics in machine learning to evaluate our model performance. In this study, positive class is represented by playing episode, while negative class is represented by non-playing episode. The metrics chosen are as follows:

1. Accuracy: The number of true positives (TP) and true negatives (TN) over the total number of observations. In our case, it indicates the percentage of both playing and non-playing episodes detected correctly.
2. Recall/Sensitivity: The number of TP over the number of all real positive observations. In our case, it indicates the number of playing episodes correctly detected over all the actual playing episodes.
3. Precision: The number of TP over all the observations that are classified as positive. In our case, it indicates the number of playing episodes correctly detected over all the episodes recognized as playing by the model.
4. Specificity: The number of true negatives (TN) over the number of all real negative observations. In our case, it indicates the number of non-playing episodes correctly detected over all the actual non-playing episodes.

Among these four metrics, accuracy is used as the primary indicator of the performance of a model. The other three metrics are used as secondary indicators in our study.

## 3. Results

### 3.1. Baselines

We selected five CNN-LSTM networks: (1) InceptionV3-LSTM [31], (2) VGG16-LSTM [34], (3) ResNet18-LSTM, (4) ResNet34-LSTM (our proposed baseline), and (5) ResNet50-LSTM [33] as our baselines for comparison. We also included the majority baseline for comparison, which in our case is a classifier that always predicts the sample as the negative class (non-playing episode). The architecture of all the CNN-LSTM baselines is the same as the original network [50], with different CNN backbone models used. All the CNN backbone models were pretrained on ImageNet.

Table 4 shows the evaluation results of the selected baselines. The reported numbers were produced by the best accuracy models identified using Test Set 1. ResNet34-LSTM performed the best on Test Set 1 in all of the metrics except for specificity. On Test Set 2, ResNet34-LSTM performed the best in all of the metrics except for recall. When the performances on Test Set 1 and Test Set 2 were both considered, the ResNet34-LSTM baseline was extended to develop our proposed variants. Table 5 presents the confusion matrices produced by the selected baseline.

**Table 4.** Baseline Evaluation Results on (**a**) Test Set 1 and (**b**) Test Set 2. Reported numbers were produced by the best accuracy models identified using Test Set 1. The best score for each metric is shown in bold. DBZ = division by zero (undefined value).

| | **(a) Test Set 1 Results** | | | | | |
|---|---|---|---|---|---|---|
| **Metrics** | **Model** | | | | | |
| | **Majority** | **InceptionV3** | **VGG16** | **ResNet18** | **ResNet34** | **ResNet50** |
| Accuracy | 0.7603 | 0.8983 | 0.7603 | 0.8923 | **0.9019** | 0.8983 |
| Recall | 0.0000 | 0.8081 | 0.0000 | 0.7980 | **0.8181** | 0.7929 |
| Precision | *DBZ* | 0.7767 | *DBZ* | 0.7633 | **0.7826** | 0.7850 |
| Specificity | 1.0000 | 0.9268 | 1.0000 | 0.9219 | 0.9283 | **0.9315** |

**Table 4.** *Cont.*

| | (b) Test Set 2 Results | | | | | |
|---|---|---|---|---|---|---|
| **Metrics** | **Model** | | | | | |
| | **Majority** | **InceptionV3** | **VGG16** | **ResNet18** | **ResNet34** | **ResNet50** |
| Accuracy | 0.7980 | 0.6897 | 0.7980 | 0.7340 | **0.7980** | 0.6995 |
| Recall | 0.0000 | 0.4390 | 0.0000 | **0.5610** | 0.4878 | 0.4634 |
| Precision | *DBZ* | 0.3103 | *DBZ* | 0.3898 | **0.5000** | 0.3275 |
| Specificity | 1.0000 | 0.7531 | 1.0000 | 0.7778 | **0.8765** | 0.7592 |

**Table 5.** Confusion Matrix Produced by ResNet34-LSTM.

| | Test Set 1 | |
|---|---|---|
| | Truth: Playful | Truth: Non-playful |
| Prediction: Playful | 162 | 45 |
| Prediction: Non-playful | 36 | 583 |
| | **Test Set 2** | |
| | Truth: Playful | Truth: Non-playful |
| Prediction: Playful | 20 | 20 |
| Prediction: Non-playful | 21 | 142 |

*3.2. Ablation Study*

We conducted an extensive ablation study on the ResNet34-LSTM network variants to examine the effectiveness of the components or modules proposed in Section 2.2. Our proposed baseline and its variants' components are depicted in Table 6. All models' performances are evaluated and compared extensively using Test Set 1 and Test Set 2.

**Table 6.** Components of the ResNet34+LSTM baseline and variants.

| Component(s) | Baseline | Variant | | | | | | | | | | |
|---|---|---|---|---|---|---|---|---|---|---|---|---|
| | | 1 | 2 | 3 | 4 | 5 | 6 | 7 | 8 | 9 | 10 | 11 |
| Flattening | ✓ | | | ✓ | | | ✓ | | | ✓ | | |
| Global average pooling | | ✓ | | | ✓ | | | ✓ | | | ✓ | |
| Global max pooling | | | ✓ | | | ✓ | | | ✓ | | | ✓ |
| Weighted average of predictions | ✓ | ✓ | ✓ | | | | | | | | | |
| Last hidden state | | | | ✓ | ✓ | ✓ | | | | | | |
| All hidden states | | | | | | | ✓ | ✓ | ✓ | ✓ | ✓ | ✓ |
| Temporal attention | | | | | | | | | | ✓ | ✓ | ✓ |

Table 7 presents the results of an ablation study conducted with the ResNet34-LSTM baseline and variants. The reported numbers were produced by the best accuracy models identified using Test Set 1. As shown in Table 7, by replacing only the baseline's flattening component with the global average pooling (Variant 1) or global maximum pooling (Variant 2) component, the variants performed better on Test Set 1. Furthermore, when only the LSTM classifier component of the baseline (i.e., weighted average of predictions) was replaced by each of the three alternative LSTM classifiers (Variant 3, Variant 6, and Variant 9), the variant of all hidden states with temporal attention provided the best accuracy among those models compared on both Test Set 1 and Test Set 2. Although our baseline gave a fairly high accuracy of 90.56% on Test Set 1, its corresponding accuracy on Test Set 2 was merely 0.99% higher than the majority baseline's accuracy of 79.80% (see Table 4).

The combination of applying global maximum pooling plus simple concatenation of all extracted spatio-temporal features (Variant 8) gave the best performance with a 92.62%

accuracy on Test Set 1, and the model performance deteriorated with a very slight difference when a temporal attention layer was added (92.13%). Based on the models' corresponding performance on Test Set 2, it can be seen that Variant 11 with global maximum pooling of spatial feature maps and temporal attention added to the concatenated spatio-temporal feature was the most robust variant. This variant had the highest accuracy (87.19%) on Test Set 2, and the second-highest accuracy (92.13%) on Test Set 1. Despite Variant 8's best performance on Test Set 1, its corresponding performance on Test Set 2 was even worse than Variant 1, which has only global average pooling applied to the extracted CNN feature maps.

In terms of the model's performance based on the value of recall, Variant 2 performed the best on Test Set 1 with 84.85%. This indicates that Variant 2 was able to identify the greatest number of actual playing episodes. On the other hand, Variant 4 had the highest precision (87.79%) and specificity (96.66%), but second-lowest recall (76.26%). This suggests that Variant 4 was able to identify the greatest number of actual non-playing episodes with the cost of the poorer ability to identify the actual playing episodes. However, both models' corresponding performances on Test Set 2 were not among the best. More specifically, for Variant 2, its accuracy on Test Set 2 was worse than the majority's baseline accuracy.

Table 8 presents the results of an ablation study in which the reported numbers were this time produced by the best accuracy models identified using Test Set 2. Here, in line with the findings from Table 7, Variant 11 performed the best in both test sets, with accuracy noticeably higher than the baselines. In addition, it is interesting to note that Variant 7's performance on Test Set 2 was perfect (100%) in precision and specificity. In other words, Variant 7 did not make any false positive predictions, and all the episodes recognized as playing episodes were the actual playing episodes. However, it had the lowest recall: it could only recognize 17.07% and 64.65% of the actual playing episodes in Test Set 2 and Test Set 1, respectively. Additionally, Variant 10 had the highest recall on both Test Set 1 and Test Set 2 but with the cost of relatively low precision. Overall, all the models with high accuracy also came with reasonable recall, precision, and specificity, suggesting that accuracy is a good indicator to evaluate a classification model for pig playing and non-playing episodes.

**Table 7.** Performances of the ResNet34+LSTM baseline and variants on (**a**) Test Set 1 and (**b**) Test Set 2 with different combinations of components. Reported numbers were produced by the best accuracy models identified using Test Set 1. The best score for each metric is shown in bold.

| (a) | | | | |
|---|---|---|---|---|
| **Model** | **Metrics** | | | |
| | **Accuracy** | **Recall** | **Precision** | **Specificity** |
| Baseline | 0.9056 | 0.8131 | 0.7970 | 0.9347 |
| Variant 1 | 0.9116 | 0.8434 | 0.7990 | 0.9331 |
| Variant 2 | 0.9153 | **0.8485** | 0.8077 | 0.9363 |
| Variant 3 | 0.8608 | 0.6616 | 0.7318 | 0.9236 |
| Variant 4 | 0.9177 | 0.7626 | **0.8779** | **0.9666** |
| Variant 5 | 0.8923 | 0.7475 | 0.7914 | 0.9379 |
| Variant 6 | 0.8935 | 0.8030 | 0.7644 | 0.9220 |
| Variant 7 | 0.9165 | 0.8081 | 0.8377 | 0.9506 |
| Variant 8 | **0.9262** | 0.8283 | 0.8586 | 0.9570 |
| Variant 9 | 0.9104 | 0.8384 | 0.7981 | 0.9331 |
| Variant 10 | 0.9153 | 0.8232 | 0.8232 | 0.9443 |
| Variant 11 | 0.9213 | 0.8232 | 0.8446 | 0.9522 |

**Table 7.** *Cont.*

| | **(b)** | | | |
|---|---|---|---|---|
| **Model** | **Metrics** | | | |
| | **Accuracy** | **Recall** | **Precision** | **Specificity** |
| Baseline | 0.8079 | 0.5610 | 0.5227 | 0.8703 |
| Variant 1 | 0.8621 | **0.6098** | 0.6757 | 0.9259 |
| Variant 2 | 0.7931 | 0.5854 | 0.4898 | 0.8457 |
| Variant 3 | 0.8030 | 0.5366 | 0.5116 | 0.8704 |
| Variant 4 | 0.8030 | 0.3902 | 0.5161 | 0.9074 |
| Variant 5 | 0.7882 | 0.4878 | 0.4762 | 0.8642 |
| Variant 6 | 0.7389 | 0.3902 | 0.3636 | 0.8272 |
| Variant 7 | 0.7882 | 0.4146 | 0.4722 | 0.8827 |
| Variant 8 | 0.8325 | 0.4146 | 0.6296 | 0.9383 |
| Variant 9 | 0.8325 | 0.4878 | 0.6061 | 0.9196 |
| Variant 10 | 0.7783 | 0.2195 | 0.4091 | 0.9198 |
| Variant 11 | **0.8719** | 0.4878 | **0.8000** | **0.9691** |

**Table 8.** Performances of the ResNet34+LSTM baseline and variants on (**a**) Test Set 2 and (**b**) Test Set 1 with different combinations of components. Reported numbers were produced by the best accuracy models identified using Test Set 2. The best score for each metric is shown in bold.

| | **(a)** | | | |
|---|---|---|---|---|
| **Model** | **Metrics** | | | |
| | **Accuracy** | **Recall** | **Precision** | **Specificity** |
| Baseline | 0.8768 | 0.6586 | 0.7105 | 0.9321 |
| Variant 1 | 0.8719 | 0.4634 | 0.8261 | 0.9753 |
| Variant 2 | 0.8571 | 0.4878 | 0.7143 | 0.9506 |
| Variant 3 | 0.8235 | 0.3902 | 0.6400 | 0.9444 |
| Variant 4 | 0.8621 | 0.5854 | 0.6857 | 0.9321 |
| Variant 5 | 0.8522 | 0.5366 | 0.6667 | 0.9321 |
| Variant 6 | 0.8177 | 0.3171 | 0.5909 | 0.9444 |
| Variant 7 | 0.8325 | 0.1707 | **1.0000** | **1.0000** |
| Variant 8 | 0.8966 | 0.6341 | 0.8125 | 0.9630 |
| Variant 9 | 0.8374 | 0.3171 | 0.7222 | 0.9691 |
| Variant 10 | 0.8374 | **0.7317** | 0.5769 | 0.8642 |
| Variant 11 | **0.9015** | 0.6341 | 0.8387 | 0.9691 |

| | **(b)** | | | |
|---|---|---|---|---|
| **Model** | **Metrics** | | | |
| | **Accuracy** | **Recall** | **Precision** | **Specificity** |
| Baseline | 0.8971 | 0.7778 | 0.7897 | 0.9347 |
| Variant 1 | 0.9080 | 0.7980 | 0.8144 | 0.9427 |
| Variant 2 | 0.8874 | 0.7626 | 0.7665 | 0.9268 |
| Variant 3 | 0.8656 | 0.7020 | 0.7277 | 0.9172 |
| Variant 4 | 0.8995 | 0.8131 | 0.7778 | 0.9268 |
| Variant 5 | 0.8874 | 0.7929 | 0.7512 | 0.9172 |
| Variant 6 | 0.8317 | 0.6919 | 0.6372 | 0.8758 |
| Variant 7 | 0.8838 | 0.6465 | 0.8312 | **0.9586** |
| Variant 8 | 0.9080 | 0.8384 | 0.7904 | 0.9299 |
| Variant 9 | 0.8995 | 0.7525 | 0.8142 | 0.9459 |
| Variant 10 | 0.8596 | **0.9242** | 0.6444 | 0.8392 |
| Variant 11 | **0.9201** | 0.8232 | **0.8402** | 0.9506 |

## 4. Discussion

### 4.1. Effect of CNN Backbone for Spatial Feature Extraction

We evaluated the performances of various CNN-LSTM baselines using different CNN backbones (Table 4). It is shown that, when the CNN-LSTM network architecture of prior work related to pig playing behavior recognition [31,33,34] was adopted, all the models performed poorly in classifying the pig playing and non-playing behaviors from our video dataset. They also performed worse than the majority baseline on Test Set 2 (see Table 4b). Because the pretrained CNN backbone models of InceptionV3 and ResNet50 used in previous studies were deeper and larger (See Table 9) than our proposed CNN backbone model, ResNet34, their notably poorer performances on Test Set 2 suggest that they captured too much noise (overfitting) in our dataset. In [31,33], the noise level in the background of their episodes was relatively lower than ours, because there were fewer pigs and the lighting in the pigpen was the same and consistent for all episodes.

**Table 9.** Number of trainable parameters of all CNN-LSTM networks used in our experiments.

| Model | Number of Trainable Parameters |
|---|---|
| InceptionV3-LSTM | 56,584,527 |
| VGG16-LSTM | 20,750,791 |
| ResNet18-LSTM | 17,212,615 |
| ResNet34-LSTM | 21,424,834 |
| ResNet50-LSTM | 47,607,495 |

It is worth noting that the precision of the VGG16-LSTM model is undefined in Test Set 1 and Test Set 2, because this baseline simply predicted all samples as the negative class. The failure of the VGG16-LSTM baseline to learn the distinctive feature between playing and non-playing behaviors was also reflected in the value of recall (0.0000) and specificity (1.0000). VGG16-LSTM baseline's poor performance could be due to its failure in capturing the higher complexity of various playing and non-playing behaviors (more complicated context) in our dataset. Unsurprisingly, by using a relatively shallower ResNet18 as a CNN backbone model, although with fairly good accuracy in classifying Test Set 1, the CNN-LSTM network performed poorly on Test Set 2. The results above suggest that to obtain an optimal CNN-LSTM network for a specific video classification task, it is important to compare and evaluate CNN backbone models with different depths and widths extensively.

### 4.2. Spatial Feature and Spatio-Temporal Feature Learning in a CNN-LSTM Network for Video Classification

As detailed in Section 2.2, we experimented with alternative combinations of (1) methods of converting spatial feature maps extracted from the last layer of ResNet34 and (2) LSTM classifier used for final prediction.

Based on the models' best performance on Test Set 1 (See Table 7a), regardless of the LSTM classifier, applying global pooling methods to the CNN feature maps produced a better result than just merely flattening the CNN feature maps in general. However, there is an exception to this finding—for the LSTM classifiers that fed the last hidden state to the final fully connected layer for classification, the variant with global average pooling (Variant 4) performed better than the one with global maximum pooling (Variant 5). A possible reason for this is that with only the final hidden state utilized, the model could predict better when the spatial feature was more generic in representing the CNN feature maps.

The best synergy was found on the variant that combines the global maximum pooling component and the LSTM classifier concatenating all hidden states for final prediction, in which it achieved the highest accuracy of 92.62%. This outcome might have occurred because the classifier was able to predict based on the most distinct feature in each feature map extracted from every frame, which is important for our dataset because our main objects are the light-colored pigs that presented highly dynamic behaviors in various backgrounds across the video frames. Nonetheless, if we look at the models' corresponding

performance on Test Set 2 (See Table 7b), except for Variant 2, Variant 8, and Variant 11, all of them performed no better than the majority class baseline. This result is in accordance with our justification of separating Test Set 2 from Test Set 1 initially, as mentioned in Section 2.1.4, in which the episodes in Test Set 2 contain more irrelevant noise (dark pixels and imbalanced pixel intensity).

While our work here focused on identifying the best alternative combinations of a CNN-LSTM network, there is a need to expand our research to make more comparisons with other deep-learning-based methods for video classification such as 3D CNN models [51–53], SlowFast networks [54], and two-stream networks [55,56] after more video data are collected. All the aforementioned DL-based methods work well on large-scale video datasets [57] that contain at least 10,000 video clips in the previous studies. At the same time, we wish to highlight the fact that the SlowFast network and two-stream network are more computationally expensive than our proposed method because they require hand-crafted features such as optical flow or training of two CNNs.

### *4.3. Effect of CNN Transfer Learning*

Table 10 depicts the performances of the ResNet34-LSTM baseline and its two variants (Variant 8 and Variant 11) on Test Set 1, with and without fine-tuning a pretrained ResNet34. All the models performed much worse when ResNet34 was used solely as a spatial feature extractor without any fine-tuning. Particularly, for Variant 11, which contains more parameters induced by the temporal attention mechanism, its performance was even worse than the majority baseline due to the severe overfitting problem, with an accuracy of 71.19% when no fine-tuning was performed.

**Table 10.** Comparison of the models' performances on Test Set 1 with and without CNN backbone fine-tuned in the training process. Reported numbers are based on the highest accuracy. FT = with fine-tuning; FE = without fine-tuning.

| Model | Metrics | | | |
|---|---|---|---|---|
| | Accuracy | Recall | Precision | Specificity |
| Baseline (FT) | 0.9056 | 0.8131 | 0.7970 | 0.9347 |
| Baseline (FE) | 0.8462 | 0.5960 | 0.7152 | 0.9252 |
| Variant 8 (FT) | 0.9262 | 0.8283 | 0.8586 | 0.9570 |
| Variant 8 (FE) | 0.8136 | 0.4545 | 0.6618 | 0.9268 |
| Variant 11 (FT) | 0.9213 | 0.8232 | 0.8446 | 0.9522 |
| Variant 11 (FE) | 0.7119 | 0.6768 | 0.4351 | 0.7229 |

Figure 6 shows the localization map from the final layer of ResNet34 for a playing (object play) episode. It can be seen that without fine-tuning the ResNet34, the important region, i.e., the head of the target pig that shows object play behavior, could not be localized properly. Hence, it is conceivable that even when a sophisticated spatio-temporal feature learning method such as temporal attention is applied to a CNN-LSTM network, if the spatial representation is not learned properly for a new video classification task, the classification performance could drop substantially because the subsequently extracted spatio-temporal feature depends on a well-represented spatial feature.

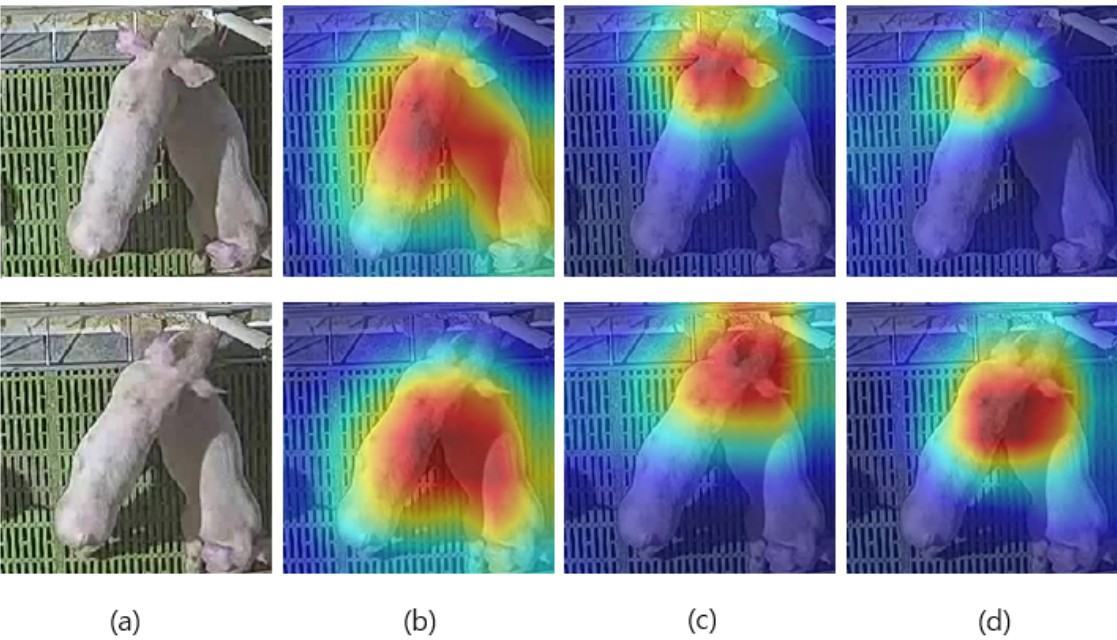

**Figure 6.** The localization map from the final layer of CNN backbone in which the important regions in the sample video frames are highlighted for prediction. The first row is the first input frame, the second row is the last input frame. (**a**) Original frame. (**b**) ResNet34 without fine-tuning. Same for the baselines and all variants. (**c**) Variant 8 with fine-tuning. (**d**) Variant 11 with fine-tuning.

*4.4. Selection of Robust Classification Model*

As highlighted in Section 2.1.4, our dataset contains a small portion of video episodes (Test Set 2) with a high level of noise, which is expected because our dataset is obtained from the real-world environment, where the videos were all recorded under a commercial pig farm setting encountered commonly in today's pig production system. In the previous studies conducted by Chen et al. [31,33,34] and Gan et al. [41], the structure of the pigpen was simpler, and the number of pigs was much lower than ours. In particular, in the work of [31,33], all the video recordings were collected from an experimental pig farm. Therefore, for our work, it is important to identify a robust model that can accurately classify the pig playing and non-playing behaviors from the video episodes and perform well against various levels of noise in the videos at the same time.

Based on the models' best performance on Test Set 2 (See Table 8a), we can see that by applying temporal attention to all hidden states, Variant 11 could attend to the spatio-temporal feature with important motion information, even when the level of noise is higher, producing the most sophisticated performance. It is thus reasonable that a variant with the best performance on Test Set 2 also performed well on Test Set 1, considering that it had learned to filter out the noise in extracting the essential spatio-temporal feature. By selecting Variant 11, which had the best performance on Test Set 2, we only need to bear a 0.61% drop in accuracy on Test Set 1. If Variant 8, which performed best on Test Set 1, is selected, we would need to sacrifice the good performance on Test Set 2, with a 6.9% drop in accuracy. Nonetheless, if all the future real-world datasets obtained are cleaner, such as Test Set 1, we would suggest to re-train Variant 8. As a side note, our observations made between Test Set 1 and Test Set 2 suggest that it would be important to install a wide-angle video camera that can cover all the corners well, minimize the number of objects that can block the camera angle, and keep the pen uncrowded in order to obtain a clean dataset in a commercial farm condition.

**5. Conclusions**

In this paper, we propose a new deep-learning-based framework designed to classify group-housed pig playing and non-playing behaviors in videos collected from a com-

mercial pig farm. We managed to develop an end-to-end trainable CNN-LSTM network without any feature engineering, using a combined architecture of ResNet34 and LSTM. We demonstrated that for our real-world video dataset that contains a wide range of dynamic pig behaviors, diversified context, and various levels of noise, the ResNet34-LSTM network provides the most robust performance when (1) global maximum pooling is applied to the spatial feature maps extracted from the last layer of the CNN backbone and (2) spatio-temporal features (hidden states) derived from each video frame are combined via temporal attention mechanism as an input to the fully connected layer for final prediction. Our work represents a substantial improvement over the existing work on automatically detecting the positive behaviors of pigs. Our proposed framework should be useful to SLF researchers and practitioners who are interested in detecting the positive behaviors of group-housed pigs and their welfare conditions.

For our future work, we aim to build the playful index per pigpen by aggregating the frequency of detected playing episodes. Given that only playing and non-playing behaviors' binary classification is considered in this work, future work should attempt to further develop a multiclass model of pig playing behaviors. Moreover, in our experiments, we manually chose Test Set 2 to test our model's robustness against the noise around the target pig in the videos due to the camera settings within a commercial pigpen. Going further, a future study might test our model's robustness against other types of noises including image processing noise such as Gaussian noise and salt-and-pepper noise. Last but not least, to build a more comprehensive group-housed pig positive welfare assessment protocol, the playing behaviors of pigs from different growth phases should also be investigated. Notwithstanding these limitations, it should also be noted that the current work has direct and useful implications on improving the current practice of SLF, while contributing to the welfare assessment of group-housed livestock animals.

**Author Contributions:** Conceptualization, B.E.L.; data curation, B.E.L.; formal analysis, B.E.L.; funding acquisition, M.Y.Y.; investigation, B.E.L. and Y.C.; methodology, B.E.L.; project administration, B.L. and M.Y.Y.; resources, B.E.L., B.L. and M.Y.Y.; software, B.E.L.; supervision, M.Y.Y.; validation, B.E.L. and Y.C.; visualization, B.E.L.; writing—original draft, B.E.L.; writing—review and editing, B.L., Y.C., and M.Y.Y. All authors have read and agreed to the published version of the manuscript.

**Funding:** This research was supported by the Korea Institute of Planning and Evaluation for Technology in Food, Agriculture and Forestry (IPET) and Korea Smart Farm R&D Foundation (KosFarm) through Smart Farm Innovation Technology Development Program, funded by Ministry of Agriculture, Food and Rural Affairs (MAFRA) and Ministry of Science and ICT (MSIT), Rural Development Administration (RDA) (421043-04-2-HD020).

**Institutional Review Board Statement:** Not applicable.

**Informed Consent Statement:** Not applicable.

**Data Availability Statement:** Restrictions apply to the availability of these data. Data were obtained from Animal Industry Data Korea (AIDK) and are available from the corresponding author with the permission of AIDK.

**Acknowledgments:** The video data used in this research were collected and processed by Animal Industry Data Korea (AIDK). The pig behavior labeling was conducted after receiving advice from the veterinarians of AIDK.

**Conflicts of Interest:** The authors declare no conflict of interest.

## Abbreviations

The following abbreviations are used in this manuscript:

| | |
|---|---|
| LSTM | Long short-term memory network |
| CNN | Convolutional neural network |
| SLF | Smart livestock farming |

## Appendix A. Ethogram

**Table A1.** The ethogram below is referred to during our data labeling process. It details out the description of pig playing behaviors for categories of object play, social play, and locomotor play.

| Play Category | Behavior | Description | Reference |
|---|---|---|---|
| Locomotor | Scamper | Two or more forward-directed hops in quick succession of each other are usually associated with excitability. | [15] |
| | Pivot | Twirling of body on the horizontal plane by a minimum of 90° is usually associated with jumping on the spot. | [15] |
| | Toss head | Energetic movements of head and neck in quick succession, in both horizontal and vertical planes. | [15] |
| | Flop | Focal animal drops to the pen floor from a normal upright position to a sitting or lying position. There is no contact with an object or another individual piglet which could cause the change in position. | [15] |
| | Hop | Focal animal has either its two front feet or all four feet off the pen floor at one time, through an energetic upwards jumping movement. The animal continues facing the same original direction for the whole of the behavior. | [15] |
| | Rolling | Lying on back, while rocking entire body in side to side movements. Behavior is terminated when focal animal returns to an upright position. | [15] |
| | Gamboling | Energetic running in forward motions within the pen environment. Normally associated with using large areas of the pen, and occasionally coming into marginal contact with other piglets (e.g., nudge). | [15] |
| Social | Pushing | Focal animal drives its head, neck or shoulders with minimal or moderate force into another piglet's body. Occasionally the behavior results in the displacement of the target piglet. | [15] |
| | Nudging | Snout of focal piglet is used to gently touch another piglet's body, with no retaliation by the recipient, excluding naso–naso contact. Usually occurs in bouts of behavior in quick succession. More intensive than mere touching, more gentle than a push. Does not include pushing past other pigs restricting passage during locomotion or joining a resting pile of piglets. | [8,13–15,22,42] |
| | Chase | Focal animal follows the locomotory movement and direction of another piglet vigorously, e.g., running after a target piglet which is also running. | [15] |
| | Push-overs | The focal animal uses its head and shoulders to drive a substantial force at a target piglet, resulting in the target to lose balance and fall over. A fall is identified by the target piglet losing its footing for at least two feet, resulting in its shoulders or hips coming into contact with the floor. | [15] |
| | Play object together | Focal animal performs object play behavior together with at least another pig on the enrichment object or toy. | [13] |
| | Play invite/Play fighting invite | Focal piglet performs locomotor or social play behaviors, which are directed through face-to-face body orientation to another non-playing piglet. The behaviors are often repeated rapidly and highly energetic. | [15] |
| | Play fighting (success) | Target piglet responds to the initiator piglets 'invite' by pushing back and engaging in a play response. Play occurs as both individuals push towards one another, with an occasional head knocking and biting attempts. | [8] |
| | Play fighting (failure) | Initiator is unsuccessful at eliciting a play response from the target individual. Target piglet either turns its head/body away from the initiator piglet, moves away without further reaction, or does not give any noticeable response to the initiator piglet's attempts to play | [8] |
| Object | Enrichment object | Manipulating the enrichment objects (toys or substrates that were deliberately put into the pen by the farmer) with mouth or snout, resulting in visible movement of the target. | [14,15] |
| | Pen facilities | Nosing or chewing any object which is part of the pen (e.g., feeder or bar of sow crate), but excluding the pen wall, floor, and enrichment object. Any behavior toward a drinking device will not be recorded. | [15] |

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
