# Peer review of "Playing Behavior Classification of Group-Housed Pigs Using a Deep CNN-LSTM Network"

_sustainability, doi:10.3390/su142316181_

Round 1

Reviewer 1 Report

This manuscript proposes a hybrid deep learning method for playing behavior classification of group-housed pigs, where CNN was used for feature extraction and LSTM was employed for classifcation task. The experimental results validated the performance of the proposed method, with satisfactory results. Overall, the topic of this research is interesting, and the manuscript was well organised and written. The detailed comments can be found as follows.

1. The contribution and innovation of the manuscript should be clarified clearly in abstract and introduction.

2. Broaden and update literature review on fundamentals of convolutional neural network or deep learning for image or data processing applications. E.g. Vision-based concrete crack detection using a hybrid framework considering noise effect. Torsional capacity evaluation of RC beams using an improved bird swarm algorithm optimised 2D convolutional neural network.

3. It is well known that the performance of deep learning model is related to the setting of hyperparameters. How did the authors set hyperparameters in this research to achieve the optimal classification performance?

4. The results of confusion matrix are suggested to be added to demonstrate the model performance.

5. In this study, the authors used different CNNs for performance evaluation. However, The advantage of the proposed method was not well presented. A comparison with other machine/deep learning methods (such as SVM) is suggested to be added in terms of evaluation metrics.

6. How about the robustness of the proposed method against noise effect?

7. More future research should be included in conclusion part.

Reviewer 2 Report

The paper is interesting and provides valuable evidence on the automated monitoring of playing behaviour in pigs. However, in my point of view, it still requires some revision.

Introduction

The introduction is rich in concepts and content. I would suggest a general review of the form and use of language to make it more readable, clear, and concise. I think it still needs some careful re-reading. Below are the points that, in my opinion, need more editing and clarification.

LL17-19: the sentence seems convoluted; I would suggest rephrasing it

LL28-34: I would suggest rephrasing the sentence to enhance clarity.

I also think that it would be appropriate to give more emphasis on the advantages and the importance of using indicators to evaluate pigs’ positive welfare, in order to introduce “Play behaviour”.

LL40-44: I believe that more fluency in the narrative would help the reader.

For example..”Several studies showed how the play behaviour is associated with positive welfare conditions such as low ammonia (cite) or an enriched environment (cite). ”…

L50: I think that the point that would be important to stress here is that, in contrast with the traditional video-monitoring, AI would allow performing this evaluation automatically.

L51: since it is the first time that appears in the text, please write AI in extended as well

L59-61: please, rephrase

L61-97: I think this part should definitely be summarised and simplified. The introduction should help to get into the topic and should entice the reader to go deeper and continue reading.

L105-115: I suggest summarising this part a lot and leaving the preceding sentence (before L110) that makes the objectives of the work and the uniqueness of the research clear and concise. More detailed descriptions of the approach should be kept in the materials and methods section.

Materials and Methods

L122-123: add a comma (,) after “nursery pigs” and after “old”

L123: I would suggest showing the mean and the standard deviation of the age

L120-127: add information on the type of video-camera used. Add details and information regarding the environment and the structure of the pens (floor, enrichment materials…etc).
For how many days the animals were recorded?

L128: I am not sure that “data description” is fitting with the content of the paragraph, maybe “data acquisition” would sound better?

L129: Why did you choose these time intervals and why did you choose to select 1 minute from the videos? Explain and add references.

L130: What do you mean by “normal activity time?” How were you assessing the sleeping time? Did you also record this behaviour? If yes, I would suggest adding this information to the ethogram (table A1).

L131: “Each processed clip (Figure 1) came with a JSON file, which contained the 131 growth phase, ID, and the corresponding bounding box coordinates of each of the pigs that 132 appear in the video frame”: It is not clear to me if the clip “came” with this information or if you had to label it. From here I understand that it was already in the clip, but from the following section “2.1.3” I understand the opposite.

L167: Please, describe which criteria were used by the authors to define an unpleasant reaction from the animals.

L172: I suggest changing the form into “Video clips were manually labeled following the ethogram….etc”

L181-183: It would be interesting to see also these data on the agreement/disagreement ratio between the two researchers during the labeling phase.

L187: What do you mean by “post-mortem discussion process”..?

L190-191: Again, given the importance of “playing with objects” for the animals involved in your trial I think that it would be important to give a detailed description in M&M of the environmental enrichments and objects available inside the pens.

LL203-203: I think it would be worth doing a unique sentence to avoid repetitions (see LL192-194)

LL248-253: I would suggest condensing into one sentence (I am not really sure that is relevant to keep the first one as it is: from LL248-250)

Discussion

LL436-434: What are the authors’ suggestions to reduce the level of noise of the videos in commercial farm conditions?  

Conclusions

I would suggest rephrasing conclusions, to make them in line with the paper's objective and more concise.

LL469-471: I would particularly rephrase this part: your work surely helps in the automatic detection of playing behaviors but does not contribute to associating them with positive welfare.

L481 “while contributing to the..” I would add “assessment” “..of group-housed livestock animals”

Round 2

Reviewer 1 Report

All the technical issues have been well addressed by the authors. I do not have further comments.

Reviewer 2 Report

I CONSIDER that the manuscript is ready to be published in the present version